# Sampling for Inference in Probabilistic Models with Fast Bayesian Quadrature

**Tom Gunter, Michael A. Osborne**
Engineering Science
University of Oxford
{tgunter,mosb}@robots.ox.ac.uk

**Roman Garnett**
Knowledge Discovery and Machine Learning
University of Bonn
rgarnett@uni-bonn.de

**Philipp Hennig**
MPI for Intelligent Systems
Tübingen, Germany
phennig@tuebingen.mpg.de

**Stephen J. Roberts**
Engineering Science
University of Oxford
sjrob@robots.ox.ac.uk

## Abstract

We propose a novel sampling framework for inference in probabilistic models: an active learning approach that converges more quickly (in wall-clock time) than Markov chain Monte Carlo (MCMC) benchmarks. The central challenge in probabilistic inference is numerical integration, to average over ensembles of models or unknown (hyper-)parameters (for example to compute the marginal likelihood or a partition function). MCMC has provided approaches to numerical integration that deliver state-of-the-art inference, but can suffer from sample inefficiency and poor convergence diagnostics. Bayesian quadrature techniques offer a model-based solution to such problems, but their uptake has been hindered by prohibitive computation costs. We introduce a warped model for probabilistic integrands (likelihoods) that are known to be non-negative, permitting a cheap active learning scheme to optimally select sample locations. Our algorithm is demonstrated to offer faster convergence (in seconds) relative to simple Monte Carlo and annealed importance sampling on both synthetic and real-world examples.

## 1 Introduction

Bayesian approaches to machine learning problems inevitably call for the frequent approximation of computationally intractable integrals of the form

$$Z = \langle \ell \rangle = \int \ell(\mathbf{x})\,\pi(\mathbf{x})\,\mathrm{d}\mathbf{x}, \tag{1}$$

where both the likelihood $\ell(\mathbf{x})$ and prior $\pi(\mathbf{x})$ are non-negative. Such integrals arise when marginalising over model parameters or variables, calculating predictive test likelihoods and computing model evidences. In all cases the function to be integrated—the integrand—is naturally constrained to be non-negative, as the functions being considered define probabilities.

In what follows we will primarily consider the computation of model evidence, $Z$. In this case $\ell(\mathbf{x})$ defines the unnormalised likelihood over a $D$-dimensional parameter set, $x_1, ..., x_D$, and $\pi(\mathbf{x})$ defines a prior density over $\mathbf{x}$. Many techniques exist for estimating $Z$, such as annealed importance sampling (AIS) [1], nested sampling [2], and bridge sampling [3]. These approaches are based around a core Monte Carlo estimator for the integral, and make minimal effort to exploit prior information about the likelihood surface. Monte Carlo convergence diagnostics are also unreliable for partition function estimates [4, 5, 6]. More advanced methods—e.g., AIS—also require parameter tuning, and will yield poor estimates with misspecified parameters.

The Bayesian quadrature (BQ) [7, 8, 9, 10] approach to estimating model evidence is inherently model based. That is, it involves specifying a prior distribution over likelihood functions in the form of a Gaussian process (GP) [11]. This prior may be used to encode beliefs about the likelihood surface, such as smoothness or periodicity. Given a set of samples from $\ell(\mathbf{x})$, posteriors over both the integrand and the integral may in some cases be computed analytically (see below for discussion on other generalisations). Active sampling [12] can then be used to select function evaluations so as to maximise the reduction in entropy of either the integrand or integral. Such an approach has been demonstrated to improve sample efficiency, relative to naïve randomised sampling [12].

In a big-data setting, where likelihood function evaluations are prohibitively expensive, BQ is demonstrably better than Monte Carlo approaches [10, 12]. As the cost of the likelihood decreases, however, BQ no longer achieves a higher effective sample rate per second, because the computational cost of maintaining the GP model and active sampling becomes relevant, and many Monte Carlo samples may be generated for each new BQ sample. Our goal was to develop a cheap and accurate BQ model alongside an efficient active sampling scheme, such that even for low cost likelihoods BQ would be the scheme of choice. Our contributions extend existing work in two ways:

**Square-root GP:** Foundational work [7, 8, 9, 10] on BQ employed a GP prior directly on the likelihood function, making no attempt to enforce non-negativity a priori. [12] introduced an approximate means of modelling the logarithm of the integrand with a GP. This involved making a first-order approximation to the exponential function, so as to maintain tractability of inference in the integrand model. In this work, we choose another classical transformation to preserve non-negativity—the square-root. By placing a GP prior on the square-root of the integrand, we arrive at a model which both goes some way towards dealing with the high dynamic range of most likelihoods, and enforces non-negativity without the approximations resorted to in [12].

**Fast Active Sampling:** Whereas most approaches to BQ use either a randomised or fixed sampling scheme, [12] targeted the reduction in the expected variance of $Z$. Here, we sample where the expected posterior variance of the integrand after the quadratic transform is at a maximum. This is a cheap way of balancing exploitation of known probability mass and exploration of the space in order to approximately minimise the entropy of the integral.

We compare our approach, termed *warped sequential active Bayesian integration (*WSABI*)*, to non-negative integration with standard Monte Carlo techniques on simulated and real examples. Crucially, we make comparisons of error against ground truth *given a fixed compute budget*.

## 2 Bayesian Quadrature

Given a non analytic integral $\langle \ell \rangle := \int \ell(\mathbf{x}) \pi(\mathbf{x}) \, \mathrm{d}\mathbf{x}$ on a domain $\mathcal{X} = \mathbb{R}^D$, Bayesian quadrature is a model based approach of inferring both the functional form of the integrand and the value of the integral conditioned on a set of sample points. Typically the prior density is assumed to be a Gaussian, $\pi(\mathbf{x}) := \mathcal{N}(\mathbf{x}; \boldsymbol{\nu}, \boldsymbol{\Lambda})$; however, via the use of an importance re-weighting trick, $q(\mathbf{x}) = (q(\mathbf{x})/\pi(\mathbf{x})) \pi(\mathbf{x})$, any prior density $q(\mathbf{x})$ may be integrated against. For clarity we will henceforth notationally consider only the $\mathcal{X} = \mathbb{R}$ case, although all results trivially extend to $\mathcal{X} = \mathbb{R}^d$.

Typically a GP prior is chosen for $\ell(x)$, although it may also be directly specified on $\ell(x)\pi(x)$. This is parameterised by a mean $\mu(x)$ and scaled Gaussian covariance $K(x, x') := \lambda^2 \exp\left(-\frac{1}{2}\frac{(x-x')^2}{\sigma^2}\right)$. The output length-scale $\lambda$ and input length-scale $\sigma$ control the standard deviation of the output and the autocorrelation range of each function evaluation respectively, and will be jointly denoted as $\theta = \{\lambda, \sigma\}$. Conditioned on samples $x_d = \{x_1, ..., x_N\}$ and associated function values $\ell(x_d)$, the posterior mean is $m_{\mathcal{D}}(x) := \mu(x) + K(x, x_d)K^{-1}(x_d, x_d)(\ell(x_d) - \mu(x_d))$, and the posterior covariance is $C_{\mathcal{D}}(x, x') := K(x, x) - K(x, x_d)K(x_d, x_d)^{-1}K(x_d, x)$, where $\mathcal{D} := \{x_d, \ell(x_d), \theta\}$. For an extensive review of the GP literature and associated identities, see [11].

When a GP prior is placed directly on the integrand in this manner, the posterior mean and variance of the integral can be derived analytically through the use of Gaussian identities, as in [10]. This is because the integration is a linear projection of the function posterior onto $\pi(x)$, and joint Gaussianity is preserved through any arbitrary affine transformation. The mean and variance estimate of the integral are given as follows: $\mathbb{E}_{\ell|\mathcal{D}}[\langle \ell \rangle] = \int m_{\mathcal{D}}(x) \, \pi(x) \, \mathrm{d}x$ (2), and

$\mathbb{V}_{\ell|\mathcal{D}}\big[\langle\ell\rangle\big] = \iint C_{\mathcal{D}}(x, x')\,\pi(x)\,\mathrm{d}x\,\pi(x')\,\mathrm{d}x'$ (3). Both mean and variance are analytic when $\pi(x)$ is Gaussian, a mixture of Gaussians, or a polynomial (amongst other functional forms).

If the GP prior is placed directly on the likelihood in the style of traditional Bayes–Hermite quadrature, the optimal point to add a sample (from an information gain perspective) is dependent only on $x_d$—the locations of the previously sampled points. This means that given a budget of $N$ samples, the most informative set of function evaluations is a design that can be pre-computed, completely uninfluenced by any information gleaned from function values [13]. In [12], where the log-likelihood is modelled by a GP, a dependency is introduced between the uncertainty over the function at any point and the function value at that point. This means that the optimal sample placement is now directly influenced by the obtained function values.

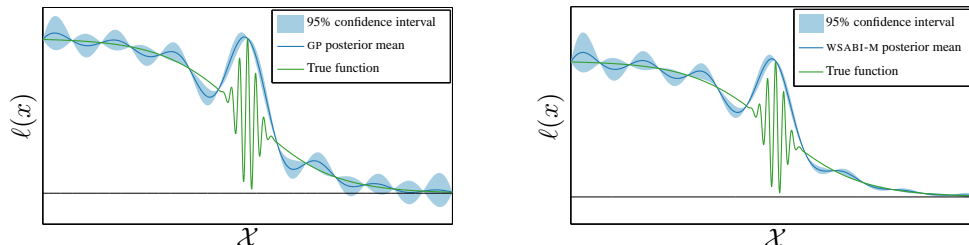

(a) Traditional Bayes–Hermite quadrature.  (b) Square-root moment-matched Bayesian quadrature.

Figure 1: Figure 1a depicts the integrand as modelled directly by a GP, conditioned on 15 samples selected on a grid over the domain. Figure 1b shows the moment matched approximation—note the larger relative posterior variance in areas where the function is high. The linearised square-root GP performed identically on this example, and is not shown.

An illustration of Bayes–Hermite quadrature is given in Figure 1a. Conditioned on a grid of 15 samples, it is visible that any sample located equidistant from two others is equally informative in reducing our uncertainty about $\ell(x)$. As the dimensionality of the space increases, exploration can be increasingly difficult due to the curse of dimensionality. A better designed BQ strategy would create a dependency structure between function value and informativeness of sample, in such a way as to appropriately express prior bias towards exploitation of existing probability mass.

## 3 Square-Root Bayesian Quadrature

Crucially, likelihoods are non-negative, a fact neglected by traditional Bayes–Hermite quadrature. In [12] the logarithm of the likelihood was modelled, and approximate the posterior of the integral, via a linearisation trick. We choose a different member of the power transform family—the square-root.

The square-root transform halves the dynamic range of the function we model. This helps deal with the large variations in likelihood observed in a typical model, and has the added benefit of extending the autocorrelation range (or the input length-scale) of the GP, yielding improved predictive power when extrapolating away from existing sample points.

Let $\tilde{\ell}(x) := \sqrt{2\big(\ell(x) - \alpha\big)}$, such that $\ell(x) = \alpha + \tfrac{1}{2}\tilde{\ell}(x)^2$, where $\alpha$ is a small positive scalar.[1] We then take a GP prior on $\tilde{\ell}(x)$: $\tilde{\ell} \sim \mathcal{GP}(0, K)$. We can then write the posterior for $\tilde{\ell}$ as

$$p(\tilde{\ell} \mid \mathcal{D}) = \mathcal{GP}\big(\tilde{\ell}; \tilde{m}_{\mathcal{D}}(\cdot), \tilde{C}_{\mathcal{D}}(\cdot, \cdot)\big); \tag{4}$$

$$\tilde{m}_{\mathcal{D}}(x) := K(x, x_d) K(x_d, x_d)^{-1} \tilde{\ell}(x_d); \tag{5}$$

$$\tilde{C}_{\mathcal{D}}(x, x') := K(x, x') - K(x, x_d) K(x_d, x_d)^{-1} K(x_d, x'). \tag{6}$$

The square-root transformation renders analysis intractable with this GP: we arrive at a process whose marginal distribution for any $\ell(x)$ is a non-central $\chi^2$ (with one degree of freedom). Given this process, the posterior for our integral is not closed-form. We now describe two alternative approximation schemes to resolve this problem.

### 3.1 Linearisation

We firstly consider a local linearisation of the transform $f \colon \tilde{\ell} \mapsto \ell = \alpha + \frac{1}{2}\tilde{\ell}^2$. As GPs are closed under linear transformations, this linearisation will ensure that we arrive at a GP for $\ell$ given our existing GP on $\tilde{\ell}$. Generically, if we linearise around $\tilde{\ell}_0$, we have $\ell \simeq f(\tilde{\ell}_0) + f'(\tilde{\ell}_0)(\tilde{\ell} - \tilde{\ell}_0)$. Note that $f'(\tilde{\ell}) = \tilde{\ell}$: this simple gradient is a further motivation for our transform, as described further in Section 3.3. We choose $\tilde{\ell}_0 = \tilde{m}_{\mathcal{D}}$; this represents the mode of $p(\tilde{\ell} \mid \mathcal{D})$. Hence we arrive at

$$\ell(x) \simeq \left(\alpha + \tfrac{1}{2}\tilde{m}_{\mathcal{D}}(x)^2\right) + \tilde{m}_{\mathcal{D}}(x)\left(\tilde{\ell}(x) - \tilde{m}_{\mathcal{D}}(x)\right) = \alpha - \tfrac{1}{2}\tilde{m}_{\mathcal{D}}(x)^2 + \tilde{m}_{\mathcal{D}}(x)\,\tilde{\ell}(x). \quad (7)$$

Under this approximation, in which $\ell$ is a simple affine transformation of $\tilde{\ell}$, we have

$$p(\ell \mid \mathcal{D}) \simeq \mathcal{GP}\big(\ell; m_{\mathcal{D}}^{\mathcal{L}}(\cdot), C_{\mathcal{D}}^{\mathcal{L}}(\cdot, \cdot)\big); \quad (8)$$

$$m_{\mathcal{D}}^{\mathcal{L}}(x) \coloneqq \alpha + \tfrac{1}{2}\tilde{m}_{\mathcal{D}}(x)^2; \quad (9)$$

$$C_{\mathcal{D}}^{\mathcal{L}}(x, x') \coloneqq \tilde{m}_{\mathcal{D}}(x)\tilde{C}_{\mathcal{D}}(x, x')\tilde{m}_{\mathcal{D}}(x'). \quad (10)$$

### 3.2 Moment Matching

Alternatively, we consider a moment-matching approximation: $p(\ell \mid \mathcal{D})$ is approximated as a GP with mean and covariance equal to those of the true $\chi^2$ (process) posterior. This gives $p(\ell \mid \mathcal{D}) \coloneqq \mathcal{GP}\big(\ell; m_{\mathcal{D}}^{\mathcal{M}}(\cdot), C_{\mathcal{D}}^{\mathcal{M}}(\cdot, \cdot)\big)$, where

$$m_{\mathcal{D}}^{\mathcal{M}}(x) \coloneqq \alpha + \tfrac{1}{2}\big(\tilde{m}_{\mathcal{D}}^2(x) + \tilde{C}_{\mathcal{D}}(x, x)\big); \quad (11)$$

$$C_{\mathcal{D}}^{\mathcal{M}}(x, x') \coloneqq \tfrac{1}{2}\tilde{C}_{\mathcal{D}}(x, x')^2 + \tilde{m}_{\mathcal{D}}(x)\tilde{C}_{\mathcal{D}}(x, x')\tilde{m}_{\mathcal{D}}(x'). \quad (12)$$

We will call these two approximations WSABI-L (for "linear") and WSABI-M (for "moment matched"), respectively. Figure 2 shows a comparison of the approximations on synthetic data. The likelihood function, $\ell(x)$, was defined to be $\ell(x) = \exp(-x^2)$, and is plotted in red. We placed a GP prior on $\tilde{\ell}$, and conditioned this on seven observations spanning the interval $[-2, 2]$. We then drew 50 000 samples from the true $\chi^2$ posterior on $\tilde{\ell}$ along a dense grid on the interval $[-5, 5]$ and used these to estimate the true density of $\ell(x)$, shown in blue shading. Finally, we plot the means and 95% confidence intervals for the approximate posterior. Notice that the moment matching results in a higher mean and variance far from observations, but otherwise the approximations largely agree with each other and the true density.

### 3.3 Quadrature

$\tilde{m}_{\mathcal{D}}$ and $\tilde{C}_{\mathcal{D}}$ are both mixtures of un-normalised Gaussians $K$. As such, the expressions for posterior mean and covariance under either the linearisation ($m_{\mathcal{D}}^{\mathcal{L}}$ and $C_{\mathcal{D}}^{\mathcal{L}}$, respectively) or the moment-matching approximations ($m_{\mathcal{D}}^{\mathcal{M}}$ and $C_{\mathcal{D}}^{\mathcal{M}}$, respectively) are also mixtures of un-normalised Gaussians. Substituting these expressions (under either approximation) into (2) and (3) yields closed-form expressions (omitted due to their length) for the mean and variance of the integral $\langle \ell \rangle$. This result motivated our initial choice of transform: for linearisation, for example, it was only the fact that the gradient $f'(\tilde{\ell}) = \tilde{\ell}$ that rendered the covariance in (10) a mixture of un-normalised Gaussians. The discussion that follows is equally applicable to either approximation.

It is clear that the posterior variance of the likelihood model is now a function of both the expected value of the likelihood at that point, and the distance of that sample location from the rest of $x_d$. This is visualised in Figure 1b.

Comparing Figures 1a and 1b we see that conditioned on an identical set of samples, WSABI both achieves a closer fit to the true underlying function, and associates minimal probability mass with negative function values. These are desirable properties when modelling likelihood functions—both arising from the use of the square-root transform.

## 4 Active Sampling

Given a full Bayesian model of the likelihood surface, it is natural to call on the framework of Bayesian decision theory, selecting the next function evaluation so as to optimally reduce our uncer-

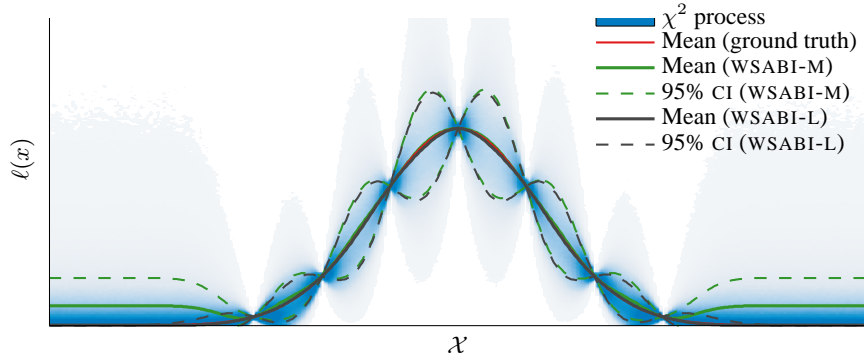

Figure 2: The $\chi^2$ process, alongside moment matched (WSABI-M) and linearised approximations (WSABI-L). Notice that the WSABI-L mean is nearly identical to the ground truth.

tainty about either the total integrand surface or the integral. Let us define this next sample location to be $x_*$, and the associated likelihood to be $\ell_* := \ell(x_*)$. Two utility functions immediately present themselves as natural choices, which we consider below. Both options are appropriate for either of the approximations to $p(\ell)$ described above.

## 4.1 Minimizing expected entropy

One possibility would be to follow [12] in minimising the expected entropy of the integral, by selecting $x_* = \arg\min_x \langle \mathbb{V}_{\ell|\mathcal{D},\ell(x)}[\langle\ell\rangle] \rangle$, where

$$\left\langle \mathbb{V}_{\ell|\mathcal{D},\ell(x)}[\langle\ell\rangle] \right\rangle = \int \mathbb{V}_{\ell|\mathcal{D},\ell(x)}[\langle\ell\rangle] \mathcal{N}(\ell(x); m_{\mathcal{D}}(x), C_{\mathcal{D}}(x,x)) \mathrm{d}\ell(x). \tag{13}$$

## 4.2 Uncertainty sampling

Alternatively, we can target the reduction in entropy of the total integrand $\ell(x)\pi(x)$ instead, by targeting $x_* = \arg\max_x \mathbb{V}_{\ell|\mathcal{D}}[\ell(x)\pi(x)]$ (this is known as *uncertainty sampling*), where

$$\mathbb{V}_{\ell|\mathcal{D}}^{\mathcal{M}}[\ell(x)\pi(x)] = \pi(x)C_{\mathcal{D}}(x,x)\pi(x) = \pi(x)^2 \tilde{C}_{\mathcal{D}}(x,x)\big(1/2\,\tilde{C}_{\mathcal{D}}(x,x) + \tilde{m}_{\mathcal{D}}(x)^2\big), \tag{14}$$

in the case of our moment matched approximation, and, under the linearisation approximation,

$$\mathbb{V}_{\ell|\mathcal{D}}^{\mathcal{L}}[\ell(x)\pi(x)] = \pi(x)^2 \tilde{C}_{\mathcal{D}}(x,x)\tilde{m}_{\mathcal{D}}(x)^2. \tag{15}$$

The uncertainty sampling option reduces the entropy of our GP approximation to $p(\ell)$ rather than the true (intractable) distribution. The computation of either (14) or (15) is considerably cheaper and more numerically stable than that of (13). Notice that as our model builds in greater uncertainty in the likelihood where it is high, it will naturally balance sampling in entirely unexplored regions against sampling in regions where the likelihood is expected to be high. Our model (the square-root transform) is more suited to the use of uncertainty sampling than the model taken in [12]. This is because the approximation to the posterior variance is typically poorer for the extreme log-transform than for the milder square-root transform. This means that, although the log-transform would achieve greater reduction in dynamic range than any power transform, it would also introduce the most error in approximating the posterior predictive variance of $\ell(x)$. Hence, on balance, we consider the square-root transform superior for our sampling scheme.

Figures 3–4 illustrate the result of square-root Bayesian quadrature, conditioned on 15 samples selected sequentially under utility functions (14) and (15) respectively. In both cases the posterior mean has not been scaled by the prior $\pi(x)$ (but the variance has). This is intended to exaggerate the contributions to the mean made by WSABI-M.

A good posterior estimate of the integral has been achieved, and this set of samples is more informative than a grid under the utility function of minimising the integral error. In all active-learning

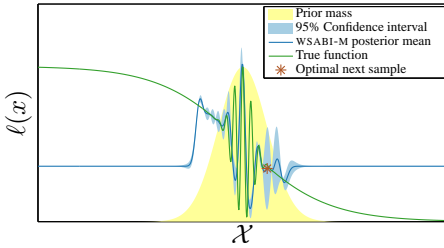

Figure 3: Square-root Bayesian quadrature with active sampling according to utility function (14) and corresponding moment-matched model. Note the non-zero expected mean everywhere.

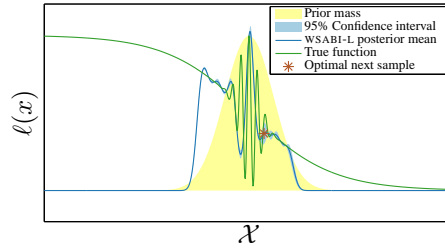

Figure 4: Square-root Bayesian quadrature with active sampling according to utility function (15) and corresponding linearised model. Note the zero expected mean away from samples.

examples a covariance matrix adaptive evolution strategy (CMA-ES) [14] global optimiser was used to explore the utility function surface before selecting the next sample.

# 5 Results

Given this new model and fast active sampling scheme for likelihood surfaces, we now test for speed against standard Monte Carlo techniques on a variety of problems.

## 5.1 Synthetic Likelihoods

We generated 16 likelihoods in four-dimensional space by selecting $K$ normal distributions with $K$ drawn uniformly at random over the integers 5–14. The means were drawn uniformly at random over the inner quarter of the domain (by area), and the covariances for each were produced by scaling each axis of an isotropic Gaussian by an integer drawn uniformly at random between 21 and 29. The overall likelihood surface was then given as a mixture of these distributions, with weights given by partitioning the unit interval into $K$ segments drawn uniformly at random—'stick-breaking'. This procedure was chosen in order to generate 'lumpy' surfaces. We budgeted 500 samples for our new method per likelihood, allocating the same amount of time to simple Monte Carlo (SMC).

Naturally the computational cost per evaluation of this likelihood is effectively zero, which afforded SMC just under 86 000 samples per likelihood on average. WSABI was on average faster to converge to $10^{-3}$ error (Figure 5), and it is visible in Figure 6 that the likelihood of the ground truth is larger under this model than with SMC. This concurs with the fact that a tighter bound was achieved.

## 5.2 Marginal Likelihood of GP Regression

As an initial exploration into the performance of our approach on real data, we fitted a Gaussian process regression model to the *yacht hydrodynamics* benchmark dataset [15]. This has a six-dimensional input space corresponding to different properties of a boat hull, and a one-dimensional output corresponding to drag coefficient. The dataset has 308 examples, and using a squared exponential ARD covariance function a single evaluation of the likelihood takes approximately 0.003 seconds.

Marginalising over the hyperparameters of this model is an eight-dimensional non-analytic integral. Specifically, the hyperparameters were: an output length-scale, six input length-scales, and an output noise variance. We used a zero-mean isotropic Gaussian prior over the hyperparameters in log space with variance of 4. We obtained ground truth through exhaustive SMC sampling, and budgeted 1 250 samples for WSABI. The same amount of compute-time was then afforded to SMC, AIS (which was implemented with a Metropolis–Hastings sampler), and Bayesian Monte Carlo (BMC). SMC achieved approximately 375 000 samples in the same amount of time. We ran AIS in 10 steps, spaced on a log-scale over the number of iterations, hence the AIS plot is more granular than the others (and does not begin at 0). The 'hottest' proposal distribution for AIS was a Gaussian centered on the prior mean, with variance tuned down from a maximum of the prior variance.

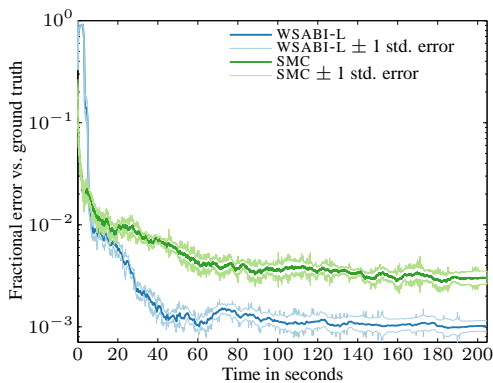

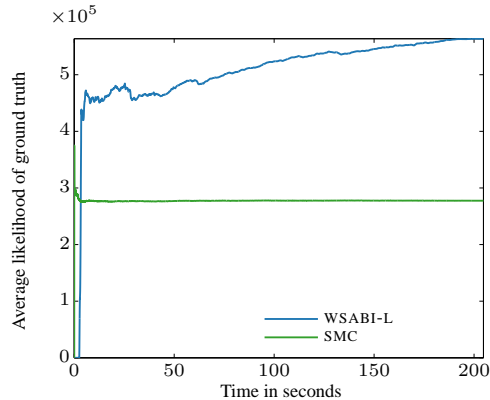

Figure 5: Time in seconds vs. average fractional error compared to the ground truth integral, as well as empirical standard error bounds, derived from the variance over the 16 runs. WSABI-M performed slightly better.

Figure 6: Time in seconds versus average likelihood of the ground truth integral over 16 runs. WSABI-M has a significantly larger variance estimate for the integral as compared to WSABI-L.

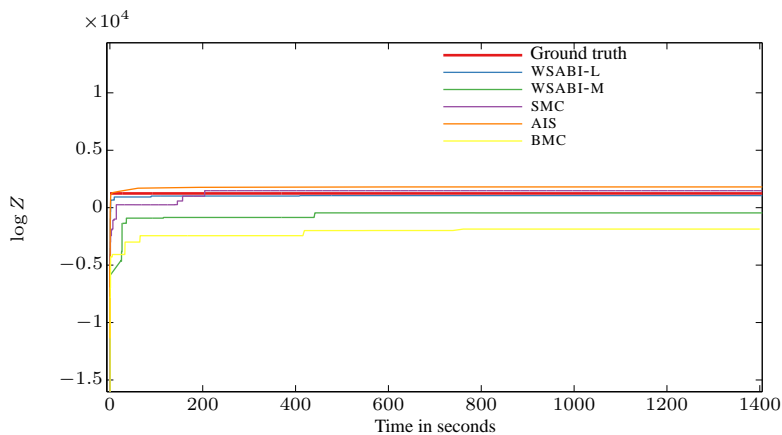

Figure 7: Log-marginal likelihood of GP regression on the yacht hydrodynamics dataset.

Figure 7 shows the speed with which WSABI converges to a value very near ground truth compared to the rest. AIS performs rather disappointingly on this problem, despite our best attempts to tune the proposal distribution to achieve higher acceptance rates.

Although the first datapoint (after 10 000 samples) is the second best performer after WSABI, further compute budget did very little to improve the final AIS estimate. BMC is by far the worst performer. This is because it has relatively few samples compared to SMC, and those samples were selected completely at random over the domain. It also uses a GP prior directly on the likelihood, which due to the large dynamic range will have a poor predictive performance.

## 5.3 Marginal Likelihood of GP Classification

We fitted a Gaussian process classification model to both a one dimensional synthetic dataset, as well as real-world binary classification problem defined on the nodes of a citation network [16]. The latter had a four-dimensional input space and 500 examples. We use a probit likelihood model, inferring the function values using a Laplace approximation. Once again we marginalised out the hyperparameters.

### 5.4 Synthetic Binary Classification Problem

We generate 500 binary class samples using a 1D input space. The GP classification scheme implemented in Gaussian Processes for Machine Learning Matlab Toolbox (GPML) [17] is employed using the inference and likelihood framework described above. We marginalised over the three-dimensional hyperparameter space of: an output length-scale, an input length-scale and a 'jitter' parameter. We again tested against BMC, AIS, SMC and, additionally, Doubly-Bayesian Quadrature (BBQ) [12]. Ground truth was found through 100 000 SMC samples.

This time the acceptance rate for AIS was significantly higher, and it is visibly converging to the ground truth in Figure 8, albeit in a more noisy fashion than the rest. WSABI-L performed particularly well, almost immediately converging to the ground truth, and reaching a tighter bound than SMC in the long run. BMC performed well on this particular example, suggesting that the active sampling approach did not buy many gains on this occasion. Despite this, the square-root approaches both converged to a more accurate solution with lower variance than BMC. This suggests that the square-root transform model generates significant added value, even without an active sampling scheme. The computational cost of selecting samples under BBQ prevents rapid convergence.

### 5.5 Real Binary Classification Problem

For our next experiment, we again used our method to calculate the model evidence of a GP model with a probit likelihood, this time on a real dataset.

The dataset, first described in [16], was a graph from a subset of the CiteSeer[x] citation network. Papers in the database were grouped based on their venue of publication, and papers from the 48 venues with the most associated publications were retained. The graph was defined by having these papers as its nodes and undirected citation relations as its edges. We designated all papers appearing in NIPS proceedings as positive observations. To generate Euclidean input vectors, the authors performed "graph principal component analysis" on this network [18]; here, we used the first four graph principal components as inputs to a GP classifier. The dataset was subsampled down to a set of 500 examples in order to generate a cheap likelihood, half of which were positive.

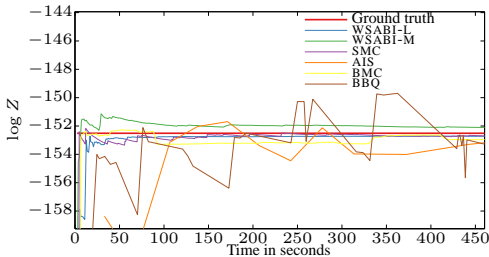

Figure 8: Log-marginal likelihood for GP classification—synthetic dataset.

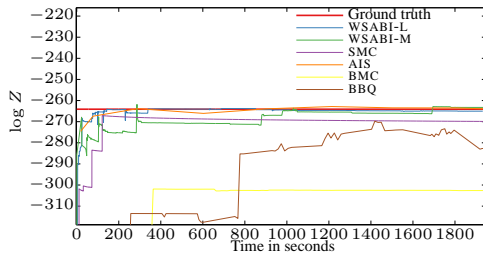

Figure 9: Log-marginal likelihood for GP classification—graph dataset.

Across all our results, it is noticeable that WSABI-M typically performs worse relative to WSABI-L as the dimensionality of the problem increases. This is due to an increased propensity for exploration as compared to WSABI-L. WSABI-L is the fastest method to converge on all test cases, apart from the synthetic mixture model surfaces where WSABI-M performed slightly better (although this was not shown in Figure 5). These results suggest that an active-sampling policy which aggressively exploits areas of probability mass before exploring further afield may be the most appropriate approach to Bayesian quadrature for real likelihoods.

## 6 Conclusions

We introduced the first fast Bayesian quadrature scheme, using a novel warped likelihood model and a novel active sampling scheme. Our method, WSABI, demonstrates faster convergence (in wall-clock time) for regression and classification benchmarks than the Monte Carlo state-of-the-art.

## Footnotes

[1]$\alpha$ was taken as $0.8 \times \min \ell(x_d)$ in all experiments; our investigations found that performance was insensitive to the choice of this parameter.

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
