[Reviews · NeurIPS 2014]

Submitted by Assigned_Reviewer_2

Overview: this paper presents a fast alternative to MC methods for approximating intractable integrals.

The main idea behind Bayesian quadrature is to exploit assumptions and regularities in the likelihood surface, something which pure Monte Carlo ignores.

The authors in this paper model the square root of the integrand (the likelihood/prior) as a Gaussian Process. Samples are then drawn according to some criterion - in this case, samples are chosen to the location of the maximal expected posterior variance of the integrand. Intuitively, this is a location where the model knows the least about the value of the integrand, and stands to gain a lot of information.

Importantly, they stress the computational benefits of their BQ active sampling method over standard Monte Carlo techniques.

The authors then approximate integrals for a variety of problems, including marginal likelihood calculation for GP regression and GP classification marginal likelihoods.

Quality -
This paper is technically sound: the problem is well motivated, the method is well described and their approach does a good job when compared to other methods for numerical integration.

Clarity-
The paper is very well written and organized. The authors do a good job conveying all aspects of the analysis. They describe Bayesian Quadrature (and numerical integration in the first place), as well as existing approaches similar to theirs in a clear way. They differentiate their own method by very clearly laying out their contributions. They do a great job explaining their approach, and the process of going from problem to solution.

Originality-
They present their method as a way to improve (both speed and some accuracy) existing methods for Bayesian quadrature. They stress two contributions: the square root GP and ‘fast active sampling’.

The square root GP seems to be another way to model a positive function (the likelihood), and one that is typically less explored. The authors also do a great job describing two ways to cope with the intractability of inference given a non-linear transformation of a GP (linearization and moment matching).

Significance-
The authors describe an alternative tool to compute a marginal likelihood - an extremely difficult and important task. The utility of such a tool is based on its speed, accuracy, and simplicity of implementation. This paper lays out an alternative solution to this common problem - one that is competitive in speed, accuracy, and simplicity. However, it remains unclear how significant this particular paper will be (or how much followup research it will inspire). What are some future directions of research made possible by this contribution?

Questions and comments:
- Figure 1: In this example, are the hyperparameters of the GP learned? If the covariance is something like a squared exponential, How does the length-scale cope with the crazy section?
- Line 208: how does the log transform compare to the square root transform. It seems somewhat clear that an unconstrained GP will more poorly, how well does another type of constrained GP perform?
- Line 264: Why is the variance of the log transform worse (worse enough to make the whole scheme worse)?
- Line 303: Is the task here just to integrate over the mixture of gaussians?
- Fig 8+9: Maybe put the converged values in a table? It’s hard to compare L, M and AIS here.

Summary: This is a good, technically sound paper describing a new method to perform Bayesian Quadrature.

Submitted by Assigned_Reviewer_8

This paper explores the application of Bayesian quadrature to Bayesian inference. This doubly Bayesian strategy was developed by Osborne et al. (NIPS 2012). The present work builds on those ideas and introduces some further innovations. The general approach is to build a probabilistic model of the posterior surface of some Bayesian inference problem, to allow ("meta") inference over quantities of interest, such as the partition function. The first innovation here is to use a squared Gaussian Process to model the likelihood (Osborne et al., used an exponentiated Gaussian Process) and the second innovation is to choose evaluate the posterior at points of high uncertainty. As before, various approximations must be made to make inference tractable.

These are valid innovations and the authors demonstrate their effectiveness. But since the obvious (and stated) precedent is that of Osborne et al., I would like to have seen "WSABI" experimentally compared against an exponentiated GP rather than just compared against the un-warped Bayesian Monte Carlo.

One pedantic point: some people find it grating to have the term "95% confidence interval" used to describe a region containing 95% of the posterior mass since it invites confusion with frequentist confidence intervals. You could consider avoiding the term. I also think the title of the paper is confusing and possibly misleading. You're using the term "sampling" in a very different sense from its usual meaning in the context of Bayesian inference. Something more explicit like "Squared Gaussian Processes for Fast Bayesian Quadrature" could be more appropriate.

Summary: Some worthwhile innovations over recent work using Bayesian quadrature for Bayesian inference.

Submitted by Assigned_Reviewer_9

The paper presents a Bayesian quadrature approach that uses square root transform and active learning technique. Two approximation schemes for the likelihood Gaussian process prior are explored -- linearization and moment matching. A simple active sampling method (without integration) is also used to increase speed.

The paper is well written. I am not an expert in BQ, so the paper is more like an educational read to me. The only part concerns me is that experimental section. The author didn't compare the method with other BQ methods. It is hard for me to judge if the method is indeed superior when it is only compared to slow MCMC methods. It would also be ideal for the authors to post the number of point it requires in order to achieve the same accuracy as MCMC methods (BMC and SMC at least).

Summary: A well written paper in BQ, but the experimental section can use some work.

Submitted by Meta_Reviewer_2

Bayesian quadrature is an interesting and important area. The proposed
method is clear and well motivated. The paper is nicely presented.
However, the empirical testing of the ideas is frustratingly weak.
Neither of the two extensions of past work (the bolded terms in the
Introduction) are actually tested directly. It could be that one or even
both of the extensions are in fact harmful.

265: "Here, on balance, we consider the square root transform
superior..." - This assertion is easily testable, but isn't tested in
the paper. I'm guessing the authors did compare to the log transform,
but I wish the paper had a report of an experiment.

423: "These results suggest that an active-sampling policy..."
- no, they don't. WSABI-L could be run without active-sampling to test
this assertion. But the only comparison is to BMC which also doesn't
have a non-linear warping.

I'm also suspicious of the Monte Carlo comparisons. These criticisms are
less important than directly testing the actual claims of the paper.

AIS is complicated, and it's fair to point out that it has many
parameters to set. However, the experiment is not reproducible and so
meaningless. How many temperatures were chosen, and how were they set?
How was the Metropolis step-size set (aiming purely for higher
acceptance rates sounds wrong). Was the step-size adjusted as a function
of temperature? (It's probably better to use an adaptive or ensemble
method as setting temperature-dependent step sizes is important but
hard.)

In Figure 7 I'm not sure whether to believe the "Ground Truth" line,
from "extensive" simple Monte Carlo sampling. The most common failure
mode for AIS is to underestimate log Z. If the chain doesn't mix well it
doesn't tend to move into high posterior regions from the prior sample,
and part of the integral is ignored. If the chain can sample from the
most significant regions, but there aren't enough samples or
temperatures, variance in Z becomes negative bias in log Z. It's
possible to overestimate log Z of course, as in a part of Figure 8, but
error bars based on the variation of the weights _usually_ cover an
answer less than the true log(Z). I'd work a bit harder to be sure of
the right answer.

In Figure 8 AIS is much worse than simple importance sampling,
suggesting time is wasted on unncessary MCMC steps and temperature
levels? Simple Monte Carlo looks similar to the proposed method. In
Figure 9 AIS is comparable to the proposed method.
Summary: The empirical testing of the ideas is frustratingly weak.
Neither of the two extensions of past work (the bolded terms in the
Introduction) are actually tested directly.
Author Feedback
Author rebuttal: We extend our thanks to all the reviewers for their helpful feedback and many thoughtful comments.

Comparison against existing BQ work
===

## Reviewer 2

- There is previous work on using the log transform (Osborne et al. 2012); however, the approximations required to yield a tractable approach are more aggressive, and, in our experience, *less reliable.*
- The resulting method is also excessively computationally expensive per sample: selecting a sample requires the costly estimation of the impact on the uncertainty in the whole integral, necessitating the maintenance of a set of candidate points in X. *This approach is hence not wall-clock competitive with MCMC or our method.*

## Reviewer 8

- Please see above.
- We do actually compare to the original BQ (Rasmussen et al.), a.k.a. ‘Bayesian Monte Carlo’ (BMC). We do not compare to the work on modelling the log of the integrand for reasons stated above.

## Reviewer 9

- Please see above.

Other
===

## Reviewer 2

- The hyperparameters of the integrand model are updated using MAP each time a new sample is selected.
- The ‘crazy’ section is actually a Gabor wavelet, so still C-inf differentiable (smooth). The method can cope with the non-stationarity in two ways -- either by shortening the length scale, (less desirable), or by maintaining a long length scale and sampling more intensively in the ‘crazy’ section. ‘Squaring’ the GP biases the naive uncertainty sampler to target areas of high expected function value as well as high uncertainty, so in reality the approach compromises between those two extremes. Realistically we expect this prior to perform well assuming the integrand is differentiable at least a few times.
- Line 303: Yes, our first test was to simply integrate over a Gaussian mixture, as we can calculate the ground truth integral analytically.
- Fig 8 + 9: We agree that we should make the results more readable -- perhaps adding a table, as you suggest.

## Reviewer 8

- If it avoids confusion with frequentist CIs, we are all for replacing “confidence intervals” with “credible intervals”.
- Regarding the title, we are not sure we follow here -- do you mean that people inevitably assume that sampling implies selecting points by constructing a Markov chain with the posterior as its equilibrium distribution? If this is true, then we imagine the title could mislead some and perhaps a rethink is required. To us, however, ‘active sampling’ is simply defined as selecting a set of points by minimising some measure of uncertainty about the model or integral. These points can then be used to compute the integral.

## Reviewer 9

- We do not follow your definition of ‘slow’ MCMC methods. We have compared the approach to the standard modern methods for calculating marginal likelihoods -- e.g. annealed importance sampling.